# Cell-Free Nucleic Acids for Early Diagnosis of Acute Ischemic Stroke: A Systematic Review and Meta-Analysis

**DOI:** 10.3390/ijms26041530

**Published:** 2025-02-12

**Authors:** Xiaodan Zhang, Yuee Cai, Brian Hon Man Sit, Rain Xiaoyu Jian, Yasine Malki, Yilin Zhang, Christopher Chi Yat Ong, Qianyun Li, Rex Pui Kin Lam, Timothy Hudson Rainer

**Affiliations:** 1Department of Emergency Medicine, School of Clinical Medicine, Li Ka Shing Faculty of Medicine, The University of Hong Kong, Hong Kong, China; u3010118@connect.hku.hk (X.Z.); evycai@hku.hk (Y.C.); briansit@connect.hku.hk (B.H.M.S.); u3569826@connect.hku.hk (R.X.J.); elaineyz@hku.hk (Y.Z.); chriso@connect.hku.hk (C.C.Y.O.); u3008308@connect.hku.hk (Q.L.); lampkrex@hku.hk (R.P.K.L.); 2Department of Chemical Pathology, The Chinese University of Hong Kong, Hong Kong, China; ymalki@link.cuhk.edu.hk

**Keywords:** blood biomarker, cell-free DNA, cell-free RNA, diagnosis, ischemic stroke

## Abstract

Rapid identification of acute ischemic stroke (AIS) is challenging in both pre-hospital and hospital settings. We aimed to identify the most promising cell-free nucleic acids (cfNAs) as diagnostic biomarkers for IS within 72 h from symptom onset. We searched PubMed, Web of Science, EMBASE, and Cochrane Library for published articles that evaluated blood cfNAs in the early diagnosis of AIS until 10 May 2023. The diagnostic performances of individual cfNAs were pooled by random-effects meta-analysis based on the fold change of biomarkers’ level between AIS and non-AIS patients. Of 2955 records, 66 articles reporting 143 different cfNAs met the inclusion criteria. The median sample size was 110, and 21.4% of the studies performed validation. Among selected high-quality studies, miR-106b-5p, miR-124, miR-155, lncRNA H19, and cfDNA showed good diagnostic performance. Data from four studies on cfDNA involving 355 AIS patients and 97 controls were pooled in the meta-analysis, which showed a significant fold change between AIS and controls (pooled ratio 1.48, 95% confidence interval 1.23–1.79, *p* < 0.001). This review highlights that cfDNA, miR-106b-5p, miR-124, miR-155, and lncRNA H19 are the most promising biomarkers for AIS diagnosis, and further research is needed for verification.

## 1. Introduction

Stroke is a prevalent cerebrovascular disease and ranks as the third-leading cause of death and the fourth-leading cause of disability worldwide [1]. The global burden of stroke is substantial, causing over 143 million healthy life years to be lost annually and imposing a substantial economic burden, accounting for 0.7% of the global gross domestic product (GDP) [2]. Ischemic stroke (IS) accounts for 65.3% of all strokes, while hemorrhagic stroke (HS) constitutes 28.8%, and subarachnoid hemorrhage constitutes 5.8% of incident strokes [1]. Currently, the treatment for acute IS (AIS) is reperfusion therapy, including intravenous thrombolysis (IVT) within 6 h, and endovascular therapy (EVT) within 24 h of symptom onset [3]. Early diagnosis of AIS and prompt initiation of reperfusion therapy can significantly reduce stroke mortality and disability [3].

However, there are unmet clinical needs for rapid diagnosis of AIS in both pre-hospital and in-hospital settings. In pre-hospital contexts, although stroke recognition scales such as the Face–Arm–Speech Test (FAST) scale aid in recognizing specific stroke symptoms, their effective application necessitates repetitive training, with stroke diagnosis often contingent on the subjective assessment of emergency responders [4]. A study highlighted that merely 25.3% of IS patients were identified and promptly transferred to a stroke center within 3 h of symptom onset, whereas approximately 41.4% reached the hospital more than a day later, indicating missed opportunities for timely reperfusion therapy [5]. In-hospital diagnosis of IS relies on clinical assessment and neuroimaging. Non-contrast computed tomography (CT) scans can rule out HS, but often fail to identify acute ischemic lesions within 12 h of onset of stroke or distinguish IS from stroke mimics (SM) [6]. While MRI is more sensitive in detecting AIS, access is limited in most emergency departments (EDs) [6]. Under current management, approximately 25% of patients who receive IVT for presumed AIS are later identified as SM, leading to inappropriate treatment [7]. Moreover, 14% of AIS cases are not recognized in the ED, resulting in missed early intervention opportunities [8].

Efforts to address these challenges have focused on developing reliable and minimally invasive biomarkers for early IS diagnosis. While downstream biomarkers, including proteins and metabolites, have been explored, none of them have been applied in clinical practice [9]. Recent advancements in high-throughput sequencing and fragmentomics have unveiled the potential of cell-free nucleic acids (cfNAs) for the diagnosis of IS, including cell-free DNA (cfDNA) and cell-free RNA (cfRNA) [10]. CfDNA, a promising domain in fluid biopsy, finds diverse clinical applications across cancer patient management and prenatal screening, predominantly centered on genetic and epigenetic anomalies in circulating nucleic acids [10]. CfRNAs, comprising mRNA and non-coding RNA (lncRNA, miRNA, tRNA, YRNA, and piRNA), are involved in epigenetic regulation and many biological processes [11].

We believe cfNAs hold promise as diagnostic markers for IS for several reasons. (a) Early detection advantage: As upstream biological markers, cfNAs can reflect changes in cellular nucleic acids earlier than proteins and metabolites during disease onset, thus providing timely insights into stroke onset. (b) Brain tissue specificity: Necrosis and apoptosis of brain cells following a stroke could trigger the rapid release of cfNAs into circulation [12,13]. Through fragment and epigenetic analyses, it may be feasible to identify cfNAs originating from brain tissue as brain-specific biomarkers [12]. (c) Disease specificity: Released due to cell death and active secretion, cfNAs participate in epigenetic regulation and pathophysiological processes, thereby directly reflecting the pathophysiological changes of cerebral infarction, uncovering the molecular essence of the disease, and supplementing information from traditional downstream markers. With the advent of point-of-care testing (POCT) technologies, such as nanopore sequencing, cfNAs can be swiftly detected [14]. By leveraging these novel POCT technologies, rapid stroke diagnosis in pre-hospital and resource-limited settings can be achieved by detecting suitable cfNAs.

While the golden reperfusion treatment window for AIS patients is within 24 h from symptom onset, the window for EVT is expanding. Mounting evidence from small observational cohort studies suggests the potential benefits of EVT beyond 24 h [15,16,17]. Noteworthy trials such as LATE-MT (Large Artery Occlusion Treated in Extended Time with Mechanical Thrombectomy, NCT05326932) and BAOCHE2 (Basilar Artery Occlusion Chinese Endovascular Trial in the Extended Time Window, NCT06560203) are currently ongoing to assess the efficacy of EVT in selected AIS cases presenting within 24 to 72 h of symptom onset.

The current research gaps are: (a) CfNAs possess significant potential for IS diagnosis; however, there is no existing systematic review that has compiled the relevant evidence; (b) previous systematic reviews mostly focused on stroke biomarkers within 24 h of symptom onset [9,18]. Given the future possibility of extending the reperfusion time window to 72 h, there is a need to identify promising cfNAs that cover such an extended time window. In this systematic review and meta-analysis, we aimed to summarize and assess the diagnostic performance of cfNAs in blood for early diagnosis of AIS within 72 h of symptom onset.

## 2. Methods

### 2.1. Search Strategy

We performed a systematic review and meta-analysis in compliance with the Preferred Reporting Items for Systematic Reviews and Meta-analyses (PRISMA) reporting guidelines [19]. Following the PRISMA and the Cochrane Handbook, the most crucial data sources are CENTRAL accessible via the Cochrane Library, MEDLINE available through PubMed, and EMBASE [19,20]. PubMed encompasses more than 18 million citations sourced from the MEDLINE database, focusing on life sciences and biomedical topics [20]. The Cochrane Library houses bibliographic reports of clinical controlled trials [20]. EMBASE, a prominent biomedical database, contains over 32 million records [20]. On the advice of the librarian at the University of Hong Kong, we conducted an expanded search on the Web of Science. Web of Science provides citation data from academic journals and conference proceedings in various academic disciplines [20]. We searched PubMed (1971–), Web of Science (1900–), EMBASE (1974–), and Cochrane Library (1973–) for articles published in English and Chinese, which reported data from studies that evaluated cfNAs as diagnostic biomarkers for IS. Our search was conducted on 10 May 2023, using the following search terms: “cell free nucleic acids”, “plasma”, “serum”, “blood”, “stroke”, “diagnosis”, “prognosis”, and their corresponding synonyms. The full list of search terms is included in Appendix A. Bibliographies of the included articles were also searched to identify additional studies that were eligible.

### 2.2. Inclusion and Exclusion Criteria

Studies were included if: (1) studies reported levels of cfNAs in patients ofAIS compared with non-AIS groups (HS, transient ischemic attack (TIA), SM, risk control (RC), and healthy control (HC)); (2) retrospective, prospective, and case-control studies without any restrictions on publication date; (3) cfNAs were tested in blood within 72 h after symptom onset. The exclusion criteria were: (1) blood samples collected > 72 h after symptom onset; (2) letters to editors, comments, conference abstracts, and case reports; (3) studies with only animal or cell culture data; (4) subjects under 18 years of age; (5) cfNAs tested in samples other than blood, e.g., cerebrospinal fluid, urine, and saliva; (6) articles written in languages other than English or Chinese.

### 2.3. Study Screening

The study protocol was prospectively registered on PROSPERO (ID: CRD42023450936). Covidence systematic review software (Veritas Health Innovation, Melbourne, Australia, www.covidence.org, assessed on 11 May 2023) was used to perform title and abstract screening, full-text review, data extraction, and quality assessment. Two reviewers (Q.L. and X.Z.) independently performed title and abstract screening. Disagreements were resolved by discussion between them. We resorted to the third reviewer for unresolved conflicts (T.R.). X.J. and X.Z. independently performed full-text screening, with disagreements resolved by discussion with the third reviewer (T.R.).

### 2.4. Study Selection

A total of 2955 records were initially identified from the following databases: PubMed (*n* = 790), Web of Science (*n* = 608), EMBASE (*n* = 1110), Cochrane Library (*n* = 413), and bibliographic search (*n* = 34). After removing duplicates, the title and abstract of 2068 articles were screened. We excluded 1843 irrelevant articles and 36 conferences/abstracts. Of the remaining 189 articles read in full text, 66 papers were included. Figure 1 depicts the selection process based on the PRISMA flow diagram.

### 2.5. Data Extraction

Data were independently extracted by two authors (H.M. and X.Z.), and disagreements between individual judgments were resolved through discussion with the third reviewer (T.R.). The missing or unreported data were requested from the corresponding authors of the respective articles by one-round email. For studies including both AIS and HS, AIS data were extracted. For studies reporting more than one biomarker, data on each biomarker was extracted separately. For studies reporting panels of biomarkers, each panel of biomarkers was considered as a single entity.

The following information was extracted from each included study: (1) bibliographic details, including the first author, study title, publication year, country; (2) study characteristics, including study design, sample size, geographical location, validation or replication methods; (3) demographics and clinical information, including age, sex, specimen, study groups, time interval between symptom onset and blood sampling; (4) biomarker measurements, including type of sample (e.g., whole blood, serum, and exosome), type of biomarkers, and their measured values expressed as means with standard deviations (SDs) or medians with interquartile ranges (IQRs). If the biomarker measurement values were not reported, we used plotdigitizer (https://plotdigitizer.com/app, accessed on 1 September 2023) to extract mean (SD) or median (IQR) values from the corresponding figures. The hazard ratio (HR), odds ratio (OR), area under the curve (AUC), sensitivity, specificity, positive predictive value (PPV), negative predictive value (NPV), false positive rate, and cutoff value, if available, were extracted or manually calculated. The adjustment for confounders was extracted as reported.

During the data extraction process, we noticed that different studies employed various extraction and sequencing methods. Although the aim of this review is not to discuss the technology, we have provided relevant comments in the discussion section.

Data were extracted using Covidence (https://www.covidence.org/, accessed on 1 September 2023) and exported in a CSV spreadsheet. Two reviewers (E.C. and X.Z.) double-checked all the extracted data.

### 2.6. Quality Assessment

The quality assessment of all the included studies was performed independently by two authors (H.S. and X.Z) using the Quality Assessment of Diagnostic Accuracy Studies-2 (QUADAS-2) [21]. Any disagreement was resolved by consulting with the third reviewer (T.R.).

### 2.7. Statistical Analysis

We first evaluated the clinical and methodological heterogeneity of the included studies. A meta-analysis was performed since there were more than three studies with acceptable clinical and methodological heterogeneity that reported the mean (SD/SEM) or median (IQR) of the biomarker concentrations. We performed a meta-analysis using the random-effects inverse-variance method based on the ratio of mean (ROM), which is IS to control ratios of mean biomarker concentration; each specific ROM was generated within a single study. The median (IQR) was converted to mean (SD) [22,23], and then the ratio of means and 95% confidence interval (CI) were calculated [24,25]. A ratio above one indicates that the concentration of the biomarker is higher in the IS group than in the control group, and a ratio below one indicates the concentration is higher in the control group. We used the I^2^ statistic to quantify the statistical heterogeneity, with an I^2^ > 75% considered substantially heterogeneous. We also considered the *p*-value from the chi-squared test to assess if this heterogeneity was significant (*p* < 0.1). Forest plots were produced to display the direction and magnitude of effects and the degree of overlap among the CIs. Publication bias was assessed with funnel plots (Appendix A). The asymmetry of the funnel plot will be evaluated through both visual assessment and the Egger test. A statistically significant reporting bias will be considered when the *p*-value is less than 0.05 in the Egger test. All analyses were performed using R version 4.2.2.

## 3. Results and Discussion

### 3.1. Results

#### 3.1.1. Study Characteristics

The 66 included studies involved 9496 subjects (5602 AIS patients, 3537 controls, 132 HS patients, 127 patients with TIA, and 98 patients with SM) from 16 countries or regions. Approximately 81.8% and 10.6% of studies were conducted in Asia and Europe, respectively. Out of the 27 studies that reported patient ethnicity, 88.8% included Asians, and 11.1% included Caucasians. The included studies had a median sample size of 110 patients (IQR 69–200, range 13–492), and most (72.7%) were case-control studies. Fourteen (21.2%) studies had validation cohorts, and three (4.5%) studies included both validation and replication cohorts. Approximately half of the included studies (*n* = 35) provided information on matched age and gender between the AIS and non-AIS groups. Additionally, 27 studies matched age, gender, and risk factors such as hypertension and diabetes. The detailed characteristics of the 66 included studies can be found in Appendix A.

#### 3.1.2. Quality Assessment of the Included Studies

The results of the assessment of the risk of bias and applicability using the QUADAS-2 are shown in Figure 2a and Appendix A. We evaluated the risk of bias by assessing the methods of patient selection, biomarker test methodology and interpretation, reference standard procedures, and flow and timing, which encompass descriptions of patients who did not undergo the biomarker test or reference standard, or were excluded from the analysis. The description and signaling questions of QUADAS-2 are shown in Appendix A. In the patient selection, six studies (9.1%) showed a high risk of bias due to non-consecutive recruitment of subjects, and 56.1% had unclear risk because details of patient selection were not reported. As for the other risk of bias domains, all the included studies were of low risk in the index test, and most had low risk in reference standard (95.5%) and flow and timing (77.3%). Regarding applicability, most studies showed low concern in patient selection (95.4%), index test (87.8%), and the reference standard being different from the review question (95.4%).

#### 3.1.3. Statistical Methods of the Included Studies

Only 37 studies (56.1%) evaluated the AUC, of which only 25 studies (37.9%) reported the respective 95% CIs (Table 1 and Table 2, Appendix A). While 29 studies (43.9%) reported sensitivity and specificity, and only 15 studies (22.7%) established a biomarker cutoff. Four studies reported adjusted OR values after adjustment for age and gender. A significant portion of the studies included (72.7%) solely visualized the differences in biomarker concentrations between patient groups on graphs. Only 27.3% showed the numerical values of biomarker concentrations in different patient groups. Furthermore, only 11 (16.7%) and 18 studies (27.3%) explored the relationship between markers and infarct volumes or National Institutes of Health Stroke Scale (NIHSS) scores, respectively.

#### 3.1.4. Reported cfNAs in Blood Sample for AIS Diagnosis

There were 143 cfNAs identified as diagnostic biomarkers for AIS from the included studies, comprising 116 miRNAs, 8 circRNAs, 11 lncRNAs, 3 mRNAs, 4 tRNAs, and cfDNA. Of all pooled biomarkers, 99 biomarkers were from AIS-controls, 6 from AIS-HS, 11 from AIS-SM, and 27 from AIS-TIA comparisons. Among the 60 articles that investigated cfRNA as stroke biomarkers, the majority (*n* = 33) analyzed cfRNA in plasma samples, 19 studies utilized serum samples, and eight focused on exosomes (Appendix A). Out of the six cfDNA studies, five evaluated cfDNA in plasma and one in serum (Table 1).

All studies collected blood samples up to 72 h after stroke onset, but the sampling times varied considerably: 3 h (2 studies), 6 h (8 studies), 9 h (1 study), 12 h (1 study), 24 h (36 studies), 48 h (7 studies), and 72 h (10 studies). Some studies measured the biomarker at multiple time points with serial changes demonstrated for individual cfNAs (Appendix A). For instance, miR-30a-5p was upregulated at 6 h after stroke onset and downregulated at 72 h. miR-124 was downregulated consistently at 24 h, 48 h, and 72 h after stroke onset. In contrast, tRNA derivatives were upregulated at 24 h, 48 h, and 7 days. Similarly, miR-503 was also upregulated at 6 h and 24 h. Taken together, these studies showed that different biomarkers have their own dynamic changes after stroke onset.

#### 3.1.5. Blood-Based cfDNAs

CfDNA was the most frequently studied biomarker (Table 1). All six studies were of high quality with a low risk of bias or concerns in applicability (Figure 2b). Nuclear DNA in plasma or serum was measured to reflect the global cfDNA levels, which included *TERT*, β-globin, *MT-ND2*, nucleic acid stain, and nucleic acid with low molecular weight (LMW). The included studies suggest that cfDNA could distinguish AIS from SM, HS, and controls. Levels of cfDNA consistently rose in AIS patients compared to SM and controls (Table 1, rows 1, 3–6) and were significantly lower compared to the HS group (Table 1, rows 2).

Two studies indicated a positive trend between cfDNA levels and NIHSS scores at admission (Table 1, rows 1, 4), and one study highlighted a positive correlation between cfDNA levels and infarct volume (Table 1, rows 1). These findings indicate a relationship between stroke severity and cfDNA release, suggesting that cfDNA can be used to monitor the severity of the disease in stroke patients.

The diagnostic metrics of cfDNA were evaluated in only two studies, which showed a moderate specificity ranging from 75% to 83% (Table 1, rows 1, 2), suggesting cfDNA holds the potential to rule in IS patients from HS patients and controls. Data from four studies comparing the cfDNA levels between AIS and controls were pooled for meta-analysis (Table 1, rows 3–6). These four studies involved 355 AIS patients and 97 controls. All four comparisons of the cfDNA concentration log ratios were above 0, with a pooled ratio of 1.48 (95% CI 1.23–1.79, *p* < 0.001; Figure 3), showing that cfDNA significantly differentiated AIS from controls. The funnel plot showed no publication bias (Appendix A).

#### 3.1.6. Blood-Based cfRNAs

None of the cfRNA have been reported more than twice. Fifteen biomarkers were reported twice: miR-106b-5p, miR-124, miR-155, miR-16, let-7b, let-7e, miR-124-3p, miR-27b-3p, miR-503, miR-185, miR-210, miR-125a-5p, miR-125b-5p, miR-143-3p, and lncRNA H19 (Appendix A). All other cfRNA biomarkers or panels were reported only once (Appendix A). Thirty-six studies reported the AUC values of cfRNA biomarkers for AIS diagnosis (Appendix A), and 26 studies compared the cfRNA levels between the AIS and non-AIS groups (Appendix A).

We summarized a list of high-quality studies without high risk of bias or application concerns that showed a high diagnostic value of cfRNA (AUC > 0.900) for AIS within 24 h (Figure 2b, Table 2). Among these high-quality studies, cell-free miRNAs were the most frequently studied. miR-106b-5p was examined in two studies that involved 355 subjects, which demonstrated a consistent rise in AIS patients’ plasma and serum within 24 h compared to controls (Appendix A, rows 1–3), with an AUC ranging from 0.962 (0.930–0.993) to 0.999 (0.997–1.000) (Table 2, row 1, 2). Serum miR-124 was consistently lower in AIS patients within 24 to 72 h of symptom onset compared to controls in two studies (Appendix A, rows 4, 5). In a cohort study of 216 subjects, low miR-124 levels showed high diagnostic accuracy for early diagnosis of AIS (Table 2, row 3). Within 24 h of stroke onset, its AUC was 0.953, which decreased to 0.949 at 48 h and 0.867 at 72 h (Appendix A, row 10, 13, and 36). MiR-155 was consistently elevated in plasma-derived microvesicles and serum in AIS patients within 24 h compared with controls with an AUC of 0.94 (Appendix A, rows 6, 7). Therefore, miR-106b-5p, miR-124, and miR-155 all show potential for the early diagnosis of IS. Specifically, with a sensitivity of 93.5, miR-124 holds the potential to rule out AIS patients, reduce inappropriate treatment, and with a specificity as high as 100%, miR-155 can be used to rule in IS patients, which enables the prompt transfer of these patients to a stroke center for early reperfusion therapy.

Furthermore, lncRNA H19 was the only lncRNA evaluated in two studies, which showed consistent upregulation in AIS patients’ plasma within 3 h compared with controls (Appendix A, rows 17, 18). Additionally, it is positively correlated to the NIHSS score, suggesting a correlation with stroke severity. One study reported an AUC of 0.91 and a specificity of 92% when IncRNA H19 was used in differentiating AIS from controls, indicating its potential as an early diagnostic biomarker for AIS (Table 2, row 8). IncRNA H19 can be used to rule in IS patients and monitor the severity of the disease.

miR-125a-5p, miR-125b-5p, and miR-143-3p have each been examined in two studies, which demonstrated a consistent elevation in expression levels in plasma and serum in the AIS group compared with controls or the SM group (Appendix A, rows 13–16). In a study that included 332 AIS patients and 160 controls, a panel comprising plasma miR-125a-5p, miR-125b-5p, and miR-143-3p was investigated across discovery, validation, and replication cohorts, which achieved a commendable AUC of 0.900 (sensitivity: 85.6%; specificity: 76.3%) for AIS (Table 2, row 10). So, this miRNA panel holds the potential to rule out AIS patients.

Two studies showed consistent elevation of plasma miR-16 in AIS patients compared to controls and HS (Appendix A, rows 8–12). While its diagnostic performance in distinguishing AIS from controls or HS was modest, miR-16 achieved an AUC surpassing 0.950 in differentiating anterior cerebral infarction or large artery atherosclerosis AIS from controls (Table 2, row 2). With a sensitivity of 100%, miR-16 can rule out IS due to large artery atherosclerosis, thus reducing the number of people who need to be scheduled for cranial computed tomography angiography (CTA) examinations.

Though not yet been investigated in two separate studies, plasma circFUNDC1, circPDS5B, and circCDC14A were found to be elevated in the AIS group compared with controls and positively correlated with the infarct volume across discovery, validation, and replication cohorts comprising 378 subjects (Appendix A, row 30). A panel combining these three circRNAs demonstrated an AUC of 0.875 in differentiating AIS from controls.

### 3.2. Discussion

Our systematic review of 66 studies provided a comprehensive overview of the diagnostic performance of various cfNAs in diagnosing AIS within 72 h of onset. We summarized a list of promising cfNAs, including miR-106b-5p, miR-124, miR-155, and lncRNA H19. Pooled data from four studies on cfDNA in the meta-analysis provided empirical evidence to support its role in early differentiation of AIS from controls. While those cfNAs exhibit promise as biomarkers for IS, rigorous validation is warranted before wider clinical adoption.

#### 3.2.1. Study Design of the Included Studies

The study design of the included studies reveals several critical areas for improvement in terms of study quality and applicability of biomarkers in diagnosing IS. Given that most studies involved participants from China, future research should encompass diverse ethnic backgrounds for generalizability to a broader population. The sample size of the included studies was small, and a priori sample size calculation was often lacking. Caution is thus advised when interpreting the results [42].

The absence of external validation in most studies raises concerns about the reliability and generalizability of the reported biomarkers. Incorporating validation cohorts is crucial for the credibility of findings [43]. Furthermore, while many studies focus on differentiating AIS from controls, the greatest clinical need lies in rapidly differentiating AIS from SM and TIA for decisions on reperfusion therapies [3]. Future studies should aim to identify AIS-specific biomarkers that can effectively differentiate AIS from these non-AIS groups instead of healthy controls. Regarding the timing of biomarker measurements, only a few studies explored serial changes in biomarker concentrations after stroke onset. Ideally, biomarker levels should exhibit rapid changes within 0–4.5 h post-stroke onset to facilitate the timely identification of thrombolysis [3]. However, only a few studies collected samples within the critical 4.5-h window. More extensive research is needed to investigate dynamic alterations pre- and post-stroke, emphasizing the importance of collecting the first samples within 4.5 h.

#### 3.2.2. Methodological Assessment of the Included Studies

Most of the included studies employed strict inclusion and exclusion criteria for IS, limiting real-world applicability. Furthermore, many studies recruited AIS patients non-consecutively, which might result in significant selection bias [44]. Future studies should consider relaxing the inclusion criteria in the validation or replication cohorts and consecutive subject recruitment AIS. Reporting the reference standard is crucial in diagnostic research, but a few studies did not specify the reference standards for IS diagnosis, which might potentially lead to bias [45].

Some studies did not clearly report the time interval between symptom onset and sampling, making comparison across studies not possible and clinical application difficult. We excluded 56 studies due to unknown blood sampling time after symptom onset, with most of them only reporting hospital admission to sampling time (Figure 1). For stroke patients, admission time does not equate to symptom onset time. Patients may be admitted to the hospital several days after symptoms onset. If studies enroll stroke patients based on hospital admission time, since we do not know how long the stroke has occurred, there is a possibility that the biomarkers identified from these studies are likely to be biomarkers for chronic stroke rather than acute stroke, and thus cannot guide reperfusion therapies in IS patients. Given the limited treatment time window for IS, future studies should report the precise symptom onset-to-sampling time [3]. However, in wake-up stroke patients or unconscious patients, the symptom onset time is often unknown. In current clinical practice, physicians rely on computed tomography perfusion to assess the ischemic penumbra for guiding the reperfusion treatment of IS patients with unclear onset times [46]. Future research could explore biomarkers that can differentiate between early-onset IS, and late-onset IS, which can be applied to detect early-stage IS in patients with unknown symptom onset times and enable timely administration of reperfusion therapy.

Regarding specimens, serum is not used in cell-free DNA studies as it contains newly released DNA from white blood cells and effects of the blood clotting process. For cfRNA studies, both serum and plasma have been used. Future studies comparing whether plasma or serum provides better diagnostic ability for cfRNA in AIS patients are needed. Of note, cfRNA in exosomes may be more stable than in plasma and serum [41]. However, exosome preparation is time-consuming and may not be suitable for rapid diagnostic experiments [47]. Additionally, biomarker concentrations may vary between peripheral venous and capillary blood [26]. Future research could consider validating results using capillary blood samples.

Moreover, most studies did not report the duration of sample storage. Given that cfNAs, especially cfRNA, degrade easily over time, this could introduce significant bias [48]. We recommend that future studies should provide detailed information on sample collection methods, the use of tubes to reduce nucleic acid degradation, pre-screening for hemolysis before sample use, and sample storage conditions and duration to minimize bias. To be noted, cfRNA is generally unstable and difficult to extract in abundance. Further work on better preservation methods of cfRNA, and evaluating how much starting material is needed for diagnosis is required. Further research into improved preservation techniques for cfRNA and the evaluation of the minimum required starting material for diagnosis are needed.

The included studies used different kits for nucleic acid extraction, such as QIAamp Circulating Nucleic Acid Kit (Qiagen, Hilden, Germany), miRCURY RNA Isolation Kit (Biofluids, Vedbaek, Denmark) [30], and miRNA Purification Kit (CW Biotech, Beijing, China) [49]. Despite the use of different kits for nucleic acid extraction, these commercial kits provide clear protocols for nucleic acid extraction from blood. Currently, the globally accessible technology for nucleic acid extraction still mainly relies on different types of commercial kits (e.g., Qiagen, Agilent) [50]. An automated nucleic acid extraction system has also been developed (BioRobot M48, Qiagen) for nucleic acid extraction from blood with standardized sample treatment, a low error rate, and avoidance of contamination; however, it is slightly more expensive than the manual extraction [51].

In the included studies, the nucleic acid sequencing was all performed using Illumina (San Diego, CA, USA) [29,30,52,53]. Other methods for multiple targeted nucleic acids detection include microarrays [27,34,54,55,56]. Nowadays, next-generating sequencing (NGS) technologies include Illumina, MGI, and PacBio sequencing platforms, with Illumina NGS systems remaining the dominant platform in the global NGS market [57,58]. Additionally, the Oxford Nanopore MinION enables real-time sequencing, rendering it applicable to a wide range of applications [59]. The validation of cfNAs is typically performed via PCR.

#### 3.2.3. Statistical Quality of Included Studies

Many of the included studies only report histograms without explicitly stating the distribution of levels in the IS and non-AIS groups. Only half of the studies reported AUC, and even fewer reported 95% CI, cutoff points, sensitivity, and specificity, which are essential for clinical interpretation and application of biomarkers [45]. Few studies evaluated the relationship between biomarkers and stroke severity, which is meaningful for identifying biomarkers for prognostication and decisions on thrombolytic therapy. Currently, AIS diagnosis is based on clinical diagnosis and neuroimaging, leaving significant gaps for pre-hospital diagnosis. Biomarkers may directly fill this gap without competing with current diagnostic methods. However, it is still necessary to compare the diagnostic accuracy of biomarkers with pre-hospital screening scales and CT scans, evaluating whether integrating biomarkers into the current diagnostic workflows can optimize care and save medical resources [4].

#### 3.2.4. cfDNA in AIS Diagnosis

In our systematic review, cfDNA emerges as the most extensively studied biomarker. Following a stroke, significant amounts of nuclear DNA are released into the bloodstream due to brain tissue necrosis, inflammatory response, NETosis, and the breakdown of the blood-brain barrier [12,13,60]. Levels of cfDNA consistently increased in the AIS cohort compared to SM and controls but decreased when compared to the HS group [36,38,39,40,41], which can possibly be explained by more severe tissue necrosis and apoptosis, and more pronounced blood–brain barrier disruption in HS [37]. Nevertheless, our meta-analysis findings indicate that cfDNA levels are elevated after stroke, exhibiting a modest yet significant effect size when comparing cfDNA concentrations in the AIS and control groups. Our findings suggest that global cfDNA levels may function as an indicator of extensive apoptosis of brain tissue and could potentially serve as a diagnostic marker for IS, and even characterizing the severity of stroke.

cfDNA originates from the release of nuclear or mitochondrial DNA from various pathophysiological processes such as apoptosis, necrosis, active secretion, autophagy, and NETosis. However, cfDNA fails to represent the core pathology of cerebral infarction and is not AIS-specific. Previous research indicates that trauma and infections can also lead to significant cell death, leading to the release of cfDNA into the circulation [61,62]. A recent comprehensive review further reveals that cfDNA levels are elevated in a wide spectrum of other neurological diseases, including central nervous system tumors, traumatic brain injury, Alzheimer’s disease, epilepsy, multiple sclerosis, and neuroinfectious diseases [12]. Therefore, it is challenging to distinguish AIS from brain trauma and other neurological diseases that rapidly induce extensive brain tissue death solely based on cfDNA.

Future studies on fragment analysis and epigenetic signatures of cfDNA could potentially indicate the tissue or cellular origin, such as brain tissue. This technology might aid in the discovery of brain-specific cfDNA for brain tissue injury assessment in stroke patients [12]. However, even if a brain-specific cfDNA is identified or cfDNA that characterizes the pathophysiological processes of cerebral infarction is found, using cfDNA as an AIS diagnostic marker remains challenging due to its low concentration in blood, dilution from higher abundance genomic DNA and high fragmentation [63]. Proper handling of blood samples is essential to prevent cell apoptosis or lysis, which can alter cfDNA levels. Reducing cellular DNA contamination and lysis can be achieved by using PAXgene Blood ccfDNA Tubes and Sterck tubes for collection, conducting secondary centrifugation, and performing pre-screening for hemolysis [64].

#### 3.2.5. cfRNA in AIS Diagnosis

cfRNAs can originate from cell death and cellular excretory processes. Non-coding RNAs, such as miRNAs, exhibit high stability in plasma and hold significant promise in diagnosing cerebral infarction. Among cfRNAs, miRNAs are the most extensively researched. Notably, miR-106b-5p [26,65], miR-124 [31,66], and miR-155 [32,67], identified in two large-sample cohorts, effectively distinguish between AIS and control groups.

The miR-106b-5p [26,65], detectable in both plasma and serum of AIS patients, shows a substantial elevation ranging from 3.63 to 23.90 folds, indicating its potential as a diagnostic biomarker for cerebral infarction. The miR-106b-5p is located on human chromosome 7q21 [68]. Functionally, miR-106b-5p enhances glutamate-induced apoptosis and induced oxidative stress and serves as a major contributor to cerebral ischemic injury [69]. A rat study suggested antagomir to miR-106b-5p ameliorate cerebral ischemia and reperfusion injury [69]. However, miR-106b-5p is not an AIS-specific biomarker; it is reported to be elevated in epilepsy [70], Alzheimer’s disease [71], and various cancers [68]. Therefore, it could be challenging to use miR-106b-5p alone to differentiate AIS from stroke mimics such as patients with epilepsy.

MiR-124 may influence AIS by modulating neuroinflammation [66]. Studies consistently report a decrease in miR-124 levels in AIS serum, with a negative correlation with infarct volume and poor outcomes, suggesting its potential as both a diagnostic and prognostic marker [31,66]. MiR-124 is a neuron-specific and brain-enriched miRNA, as it is mainly expressed in neuronal cells and is highly expressed in all brain regions [72]. After stroke onset, miR-124 could be released from damaged brain cells to circulation and serve as a diagnostic biomarker for AIS. Studies have shown that miR-124 plays a crucial role in promoting the survival and neuronal differentiation of neural stem cells. It exerts anti-apoptotic and neuroprotective effects, mitigates excitotoxicity, enhances angiogenesis, and regulates the inflammatory response [73]. As a result, miR-124 has been proposed as a potential therapeutic target after IS [73]. However, miR-124 is also related to the inflammation of neurodegenerative disorders and potentially serves as a future therapeutic biomarker [74].

Similarly, miR-155, a pro-inflammatory miRNA implicated in AIS pathophysiology, exhibits significantly elevated levels (1.7–8.5 folds) in serum and plasma-derived microvesicles of AIS patients in high-quality studies [32,67]. MiR-155 exists in different tissues, including the brain, thymus, and spleen [75,76]. MiR-155 is regarded as a pro-inflammatory regulator of the central nervous system with upregulated expression in various neurodegenerative disorders, such as multiple sclerosis and neuroviral infections [77]. Studies have shown that miR-155 is a potential protective and therapeutic target for IS. For example, miR-155 inhibitor treatment by intravenous injection in a mouse model could reduce infarct size by 34% and improve microvascular integrity as well as functional recovery [77,78,79].

Apart from miRNAs, lncRNA H19 was the only cfRNA investigated in two studies [28,80]. lncRNAs constitute a vast and predominantly unexplored functional element of the genome [81]. Numerous studies have highlighted that the brain exhibits the highest levels of lncRNA expression compared to other tissue types [81]. The expression of lncRNA H19 is globally repressed but enriched in vascular tissue [82]. Previous research has suggested that lncRNA H19 plays a role in promoting neuroinflammation and being involved in the pathophysiology of AIS [28]. Moreover, it is consistently upregulated in the AIS group within 3 h from symptom onset compared with controls, and positively correlated with stroke severity, suggesting it can serve as an acute diagnostic biomarker for IS. LncRNA H19 has played important roles in various cardiovascular and neurological pathologies, including atherosclerosis, cardiac disease (e.g., coronary artery disease and myocardial infarction), cerebrovascular disease (e.g., IS and HS), brain tumor, and neurodegenerative disease [82,83]. Therefore, lncRNA H19 also holds the potential as a diagnostic and therapeutic marker for other neurological diseases, such as brain tumors [84].

To summarize, the miR-106b-5p, miR-124, miR-155, and lncRNA H19 all hold the potential to be the diagnostic marker of AIS, but they were associated with other neurological diseases. Given the complexity of cerebral infarction, characterizing its pathophysiology solely through a single marker may be challenging. In the application of cfRNAs, combining different cfRNAs into a panel could enhance diagnostic accuracy [30]. Future studies could explore combining biomarkers representing diverse disease pathways into a panel to effectively identify AIS patients.

#### 3.2.6. POCT of cfNAs

These cfNAs, to be used as diagnostic tools, easy-to-use, efficient, and reliable POCT devices are required, and the time from blood sampling to analysis should be as short as possible. Currently, promising nucleic acid POCT technologies include the clustered-regularly-interspaced-short-palindromic-repeats (CRISPR) lateral-flow-strip tests and the single-molecule-with-a-large-transistor bioelectronic palmar devices [85]. Combining nucleic acid POCT technology with rapid cfNA separation technology can achieve rapid detection of cfNAs [86]. With the development of technologies, the sample-to-answer time is becoming shorter; for example: (i) the one-step fluorescence assay using suboptimal protospacer adjacent motifs for Cas12a in nucleic acid detection reported a sample-to-answer time of less than 20 min [87]; (ii) the electroosmotic flow-based ultrasensitive electrochemiluminescence microfluidic system can ultrasonically detect miRNAs within ten minutes [88].

#### 3.2.7. Limitations of the Systematic Review

We excluded many studies that did not report the time interval between stroke onset and sampling and did not include conference abstracts or papers, which could potentially lead to the omission of some promising cfNAs. Despite the extensive search on cfRNA, the heterogeneity of studies and the limited number of studies focusing on the same biomarkers have hindered data pooling for meta-analyses. In quality assessment, we did not consider sample handling factors, such as screening for hemolysis and sample storage duration. In the future, it may be necessary to develop specific quality evaluation scales for diagnostic biomarker studies.

## 4. Conclusions

This systematic review and meta-analysis summarize a number of differentially expressed cfNAs for AIS diagnosis and underscores the critical need for improved quality in cfNA research. It remains uncertain whether the reported biomarkers could enhance the current diagnostic approach for cerebral infarction. Nonetheless, cfDNA, miR-106b-5p, miR-124, miR-155, and lncRNA H19 emerge as the most promising AIS diagnostic markers in high-quality studies, suggesting that future prospective research could focus on validating these markers.

## Figures and Tables

**Figure 1 ijms-26-01530-f001:**
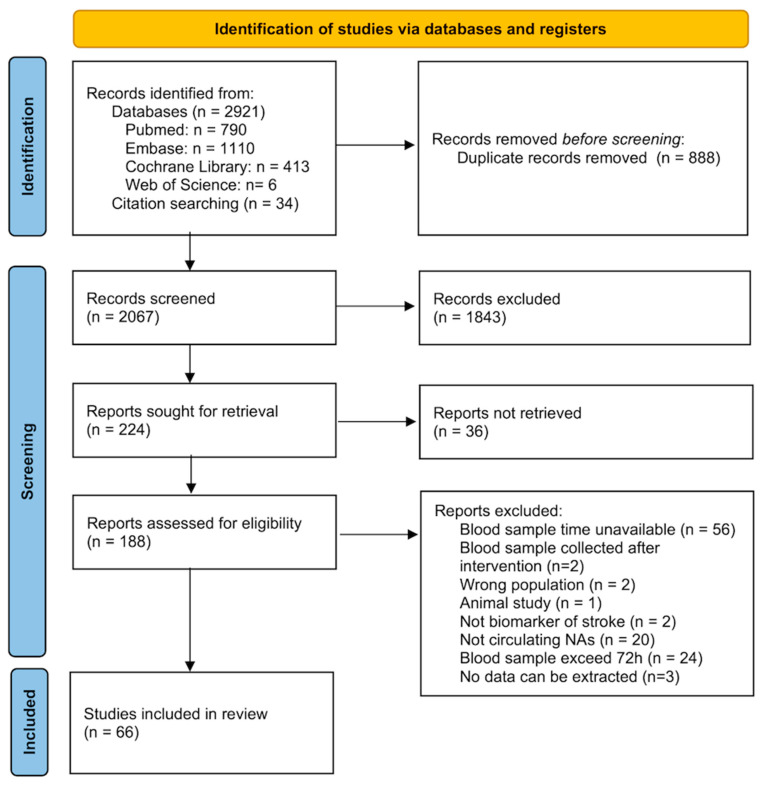
Flow diagram of study selection.

**Figure 2 ijms-26-01530-f002:**
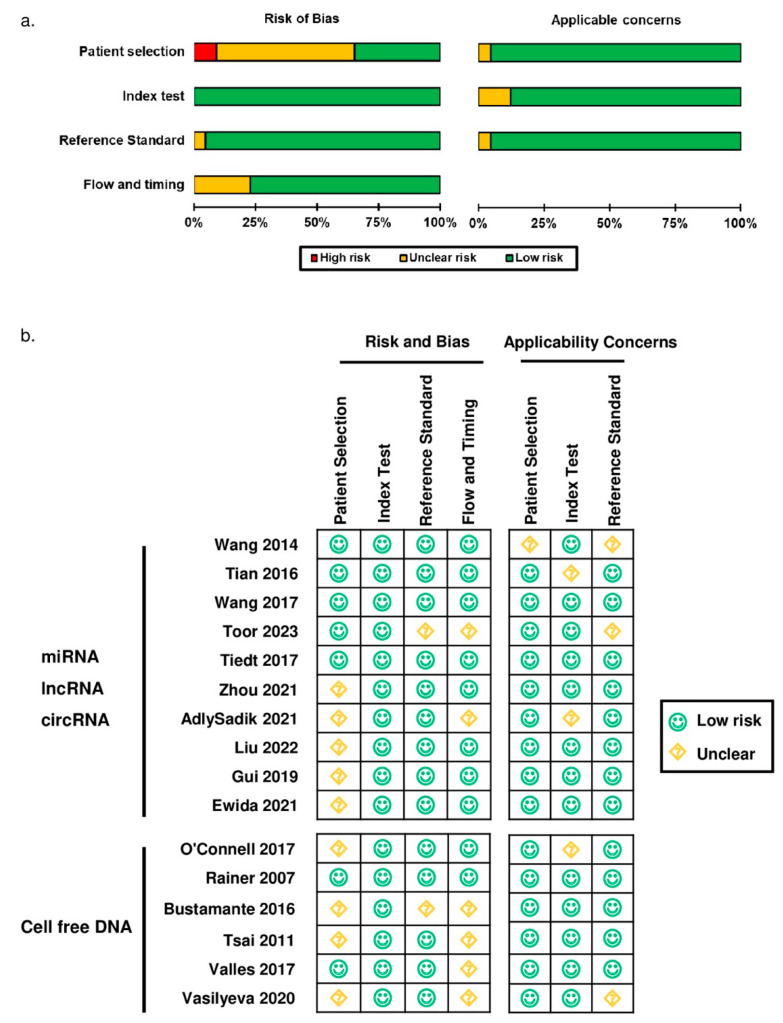
Quality assessment results using QUADAS-2 [21]. (**a**) Quality of all included studies; (**b**) high-quality cfRNA studies and all papers for cfDNA. Wang 2014 [26], Tian 2016 [27], Wang 2017 [28], Toor 2023 [29], Tiedt 2017 [30], Zhou 2021 [31], AdlySadik 2021 [32], Liu 2022 [33], Gui 2019 [34], Ewida 2021 [35], O’Connell 2017 [36], Rainer 2007 [37], Bustamante 2016 [38], Tsai 2011 [39], Valles 2017 [40], Vasilyeva 2020 [41].

**Figure 3 ijms-26-01530-f003:**
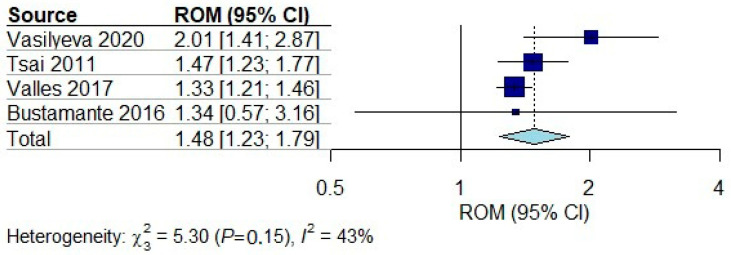
Blood ratios of cfDNA concentrations between ischemic stroke patients and controls. Individual study ratios and their corresponding 95% CIs are indicated by filled squares. The size of the square indicates the weight of the study. The pooled ratio and 95% CI are indicated by a diamond. The dotted line indicates a ratio of one. Ratio of mean (ROM). Vasilyeva 2020 [41], Tsai 2011 [39], Valles 2017 [40], Bustamante 2016 [38].

**Table 1 ijms-26-01530-t001:** The statistical information of cfDNA in the included studies.

#	Study ID	Biomarker	Comparison Group	Specimen	Sample Time	Level	AUC (95% CI)	Sen (%)	Spe (%)	Unit	Method
1	O’Connell, 2017 [36]	cfDNA (*TERT*)	IS (43) vs. SM (20)	Plasma	24 h	Up	0.857 (0.869–0.979)	86.0	75.0	Relative expression	RT-qPCR
2	Rainer, 2007 [37]	cfDNA (β-globin)	IS (118) vs. HS (35)	Plasma	24 h	Down	--	31.0	83.0	KGE/L	RT-qPCR
3	Bustamante, 2016 [38]	cfDNA (β-globin)	IS (54) vs. HC (15)	Serum	4.5 h	Up *p* > 0.05	--	--	--	KGE/L	RT-qPCR
4	Tsai, 2011 [39]	cfDNA (β-globin)	IS (50) vs. RC (50)	Plasma	48 h	Up	--	--	--	KGE/L	RT-qPCR
	Tsai, 2011 [39]	cfDNA (*MT-ND2*)	IS (50) vs. RC (50)	Plasma	48 h	Up	--	--	--	KGE/L	RT-qPCR
5	Valles, 2017 [40]	cfDNA (nucleic acid stain)	IS (243) vs. Con (27)	Plasma	72 h	Up (1.33)	--	--	--	ng/mL	Fluorescence microplate reader
6	Vasilyeva, 2020 [41]	cfDNA (LMW)	IS (8) vs. Con (5)	Plasma	3 h	Up	--	--	--	ng/mL	Isolated from gels
	Vasilyeva, 2020 [41]	cfDNA (LMW)	IS (8) vs. Con (5)	Plasma	6 h	Up	--	--	--	ng/mL	Isolated from gels
	Vasilyeva, 2020 [41]	cfDNA (LMW)	IS (8) vs. Con (5)	Plasma	12 h	Up	--	--	--	ng/mL	Isolated from gels
	Vasilyeva, 2020 [41]	cfDNA (LMW)	IS (8) vs. Con (5)	Plasma	24 h	Up	--	--	--	ng/mL	Isolated from gels
	Vasilyeva, 2020 [41]	cfDNA (LMW)	IS (8) vs. Con (5)	Plasma	48 h	Up	--	--	--	ng/mL	Isolated from gels
	Vasilyeva, 2020 [41]	cfDNA (LMW)	IS (8) vs. Con (5)	Plasma	72 h	Up	--	--	--	ng/mL	Isolated from gels

Notes: Control (Con), healthy control (HC), risk control (RC), ischemic stroke (IS), hemorrhagic stroke (HS), stroke mimics (SM), low molecular weight (LMW), National Institutes of Health Stroke Scale (NIHSS), kilogenome-equivalents/L (KGE/L). The study by O’Connell 2017 [36] showed that cfDNA (TERT) was associated with NIHSS (r = 0.246, *p* = 0.098), infarct volume (r = 0.350, *p* = 0.026) and neutrophil %WBC (r = 0.450, *p* = 0.004). The study by Rainer 2007 [37] showed that the OR of cfDNA (β-globin) adjusted for age, sex, sampling time, GCS, and NIHSS was 1.20 (1.20–1.41), while the OR of cfDNA (β-globin) adjusted for age, sex, β-globin DNA, and serum S100 was 1.23 (1.05–1.44). The study by Tsai 2011 [39] showed that cfDNA (β-globin) was associated with NIHSS (r = 0.36, *p* = 0.038) and WBC counts (r = 0.36, *p* = 0.012).

**Table 2 ijms-26-01530-t002:** The statistical information of cfRNA without high risk of bias or high application concerns in the QUADAS-2, and with an AUC > 0.900 in the included studies.

#	Study ID	Comparison Group	Specimen	Sample Time	Biomarker	Level	AUC (95% CI)	Sen (%)	Spe (%)	Normalization
1	Wang, 2014 [26]	IS (MRI-, 60) vs. Con (116)	Plasma	24 h	miR-106b-5P	Up (3.63)	0.999 (0.997–1.000)	--	--	RNU6B, 2^−ΔΔCT^
	Wang, 2014 [26]	IS (MRI+, 76) vs. Con (116)	Plasma	24 h	miR-106b-5P	Up (23.9)	0.962 (0.930–0.993)	--	--	RNU6B, 2^−ΔΔCT^
	Wang, 2014 [26]	IS (MRI-, 60) vs. Con (116)	Plasma	24 h	miR-320d	Down (0.23)	0.977 (0.952–1.000)	--	--	RNU6B, 2^−ΔΔCT^
	Wang, 2014 [26]	IS (MRI+, 76) vs. Con (116)	Plasma	24 h	miR-320d	Down (0.07)	0.987 (0.972–1.000)	--	--	RNU6B, 2^−ΔΔCT^
	Wang, 2014 [26]	IS (MRI-, 60) vs. Con (116)	Plasma	24 h	miR-320e	Down (0.33)	0.953 (0.913–0.994)	--	--	RNU6B, 2^−ΔΔCT^
	Wang, 2014 [26]	IS (MRI+, 76) vs. Con (116)	Plasma	24 h	miR-320e	Down (0.13)	0.981 (0.963–0.998)	--	--	RNU6B, 2^−ΔΔCT^
	Wang, 2014 [26]	IS (MRI+, 76) vs. Con (116)	Plasma	24 h	miR-4306	Up (5.3)	0.952 (0.922–0.982)	--	--	RNU6B, 2^−ΔΔCT^
2	Tian, 2016 [27]	IS (TACI, 4) vs. Con (23)	Plasma	6 h	miR-16	Up	0.978 (0.925–1.031)	100.0	91.3	Spiked-in cel-miR-54, 2^−ΔCT^
	Tian, 2016 [27]	IS (LAA, 9) vs. Con (23)	Plasma	6 h	miR-16	Up	0.952 (0.879–1.024)	100.0	91.3	Spiked-in cel-miR-54, 2^−ΔCT^
3	Zhou, 2021 [31]	IS (108) vs. Con (108)	Serum	24 h	miR-124	Down	0.953	93.5	91.7	U6, 2^−ΔΔCT^
4	AdlySadik, 2021 [32]	IS (46) vs. Con (50)	Serum	24 h	miR-155	Up (8.5)	0.940	85.7	100.0	U6, ΔCT
5	Liu, 2022 [33]	IS (45) vs. RC (32)	Plasma	24 h	CircOGDH	Up (54)	0.9326	82.2	96.9	GAPDH or β-actin, 2^−ΔΔCT^
6	Gui, 2019 [34]	IS (CE, 51) vs. Con (33)	Plasma	24 h	let-7e	Up	0.923 (0.859–0.998)	89.0	90.0	Spiked-in cel-miR-39, 2^−ΔΔCT^
	Gui, 2019 [34]	IS (CE, 51) vs. Con (33)	Plasma	24 h	miR-125b	Up	0.906 (0.888–0.956)	86.0	87.0	Spiked-in cel-miR-39, 2^−ΔΔCT^
7	Ewida, 2021 [35]	IS (50) vs. HS (25)	Serum	24 h	lncRNA LINK-A	Down	0.914	92.0	94.0	GAPDH, 2-ΔΔCT
8	Wang, 2017 [28]	IS (36) vs. Con (25)	Plasma	3 h	lncRNA H19	Up	0.91	80.6	92.0	β-actin
9	Toor, 2023 [29]	IS (95) vs. TIA (30)	Serum	24 h	Panels of 25 miRNA (classifier model)	Up/Down	0.901	--	--	RNA sequencing
10	Tiedt, 2017 [30]	IS (200) vs. Con (100)	Plasma	24 h	miR-125a-5p, miR-125b-5p, miR-143–3p	Up	0.900	85.6	76.3	Spiked-in UniSp2, UniSp4 and UniSp5, ΔΔCq

Notes: Control (Con), healthy control (HC), risk control (RC), stroke mimics (SM), ischemic stroke (IS), hemorrhagic stroke (HS), large-artery atherosclerosis (LAA), National Institutes of Health Stroke Scale (NIHSS), magnetic resonance imaging (MRI), total anterior circulation infarct (TACI), cardioembolic (CE), partial anterior circulation infarct (PACI), diabetes mellitus (DM), massive cerebral infarction (MCI). The study by Zhou 2021 [31] showed that CircOGDH was correlated with penumbra size (r = 0.962, *p* = 0.002). The study by Liu 2022 [33] showed that low miR-124 was correlated with poor survival. The study by Wang 2017 [28] showed that lncRNA H19 was correlated with NIHSS (3 h) (r = 0.1964, *p* = 0.0068) and NIHSS (7 d) (r = 0.6488, *p* < 0.0001). The cut point for the study of AdlySadik 2021[32] was 1.75. The cut point for the study of Ewida 2021[35] was 0.28.

## Data Availability

The original contributions presented in this study are included in the article/Appendix A. Further inquiries can be directed to the corresponding author(s).

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
