# Peer review of "Cell-Free Nucleic Acids for Early Diagnosis of Acute Ischemic Stroke: A Systematic Review and Meta-Analysis"

_ijms, 2025, doi:10.3390/ijms26041530_

Round 1
Reviewer 1 Report
Comments and Suggestions for Authors
Comments for the authors
Major corrections and suggestions
Interested to know about the timing from blood sampling to analysis, if this is to be used as a diagnostic tool, are specific cfDNAs likely to be determined by portable devices/lateral flow/like testing? Considering the wide spread of dates from which studies were collected, are these extraction methods consistent/reliable? Is there adequate and globally accessible technology today for extraction and sequencing? If this information is limited in the analysed studies, it should be included as a discussion point in the study design section.
Formatting of tables – may just be the conversion to pdf but please embed so the column width stays appropriate, very difficult to read
When discussing inclusion/exclusion criteria for studies, authors may consider that the symptom onset time is not always known, for example patients that are found discovered after a stroke, or when strokes occur at night. Some patients are not brought to hospital by paramedics. Hospital admission time is one consistent measurement that would have real-world application, and how this might effect cfDNA as a diagnostic tool.
The reader would benefit from some description of the key DNAs and RNAs mentioned, i.e cellular origin, function, prevalence/therapeutic in other neurovascular/degenerative diseases.
Minor corrections and suggestions
Line 34 - Ref 1 and 3, global burden of disease statistics were updated in 2021 - https://www.thelancet.com/journals/laneur/article/PIIS1474-4422(24)00369-7/fulltext
‘Globally, ischaemic stroke constituted 65·3% (62·4–67·7), intracerebral haemorrhage constituted 28·8% (28·3–28·8), and subarachnoid haemorrhage constituted 5·8% (5·7–6·0) of incident strokes.’
Line 263 – Figure 3 legend missing/incomplete
Line 361 – ‘we excluded 56 studies due to unknown..’
Line 367 – ‘newly released DNA’
Line 369 – ‘AIS patients’
Line 391 – missing “.” After thrombolytic therapy.
Reviewer 2 Report
Comments and Suggestions for Authors
Dear Authors
Your manuscript presents a well-structured and comprehensive systematic review on cfNAs for AIS diagnosis. However, according to my opinion, some aspects could be improved:
In your introduction, I suggest to further clarify the research gap and justify the selection of cfNAs.
-You could provide a stronger rationale for database selection and describe statistical analyses in more detail.
-You could improve the consistency of biomarker performance metrics and discuss clinical implications more explicitly (in your results)
Addressing these issues will enhance the clarity and impact of your paper.
Best Regards
